# Randomised controlled feasibility trial of an active communication education programme plus hearing aid provision versus hearing aid provision alone (ACE To HEAR)

Judith Watson ,[1] Elizabeth Coleman,[1] Cath Jackson,[2] Kerry Bell,[1] Christina Maynard,[3] Louise Hickson,[4] Anne Forster,[5] Caroline Fairhurst,[1] Catherine Hewitt,[1] Rob Gardner,[6] Kate Iley,[7] Lorraine Gailey,[8] Nicholas J Thyer [3]

For numbered affiliations see end of article.

**Correspondence to**
Dr Judith Watson;
jude.watson@york.ac.uk

**ABSTRACT:**

**Objective** To establish the acceptability and feasibility of delivering the Active Communication Education (ACE) programme to increase quality of life through improving communication and hearing aid use in the UK National Health Service.

**Design** Randomised controlled, open feasibility trial with embedded economic and process evaluations.

**Setting** Audiology departments in two hospitals in two UK cities.

**Participants** Twelve hearing aid users aged 18 years or over who reported moderate or less than moderate benefit from their new hearing aid.

**Interventions** Consenting participants (along with a significant other) were to be randomised by a remote, centralised randomisation service in groups to ACE plus treatment-as-usual (intervention group) or treatment-as-usual only (control group).

**Primary outcome measures** The primary outcomes were related to feasibility: recruitment, retention, treatment adherence and acceptability to participants and fidelity of treatment delivery.

**Secondary outcome measures** International Outcomes Inventory for Hearing Aids, Self-Assessment of Communication, EQ-5D-5L and Short-Form 36. Blinding of the participants and facilitator was not possible.

**Results** Twelve hearing aid users and six significant others consented to take part. Eight hearing aid users were randomised: four to the intervention group; and four to treatment-as-usual only. Four significant others participated alongside the randomised participants. Recruitment to the study was very low and centres only screened 466 hearing aid users over the 15-month recruitment period, compared with the approximately 3500 anticipated. Only one ACE group and one control group were formed. ACE could be delivered and appeared acceptable to participants. We were unable to robustly assess attrition and attendance rates due to the low sample size.

**Conclusions** While ACE appeared acceptable to hearing aid users and feasible to deliver, it was not feasible to identify and recruit participants struggling with their hearing aids at the 3-month posthearing aid fitting point.

**Strengths and limitations of this study**

► This is the first study to explore the potential of using Active Communication Education (ACE) as a tool to improve communication and hearing aid benefit for struggling adult hearing aid users and their significant others.

► This is the first study to identify and evaluate the processes needed to deliver an randomised controlled trial of treatment-as-usual plus ACE versus treatment-as-usual alone within audiology clinics.

► Low recruitment meant some of the trial objectives were not robustly assessed.

**Trial registration number** ISRCTN28090877.

## INTRODUCTION

Age-related hearing impairment is reported as the third most common chronic condition affecting approximately 328 million (91%) middle and older aged adults worldwide[1] and 10 million adults in the UK[2 3] with the majority having mild or worse hearing impairment, which progressively deteriorates with age.[4] Hearing loss often impacts on quality of life and general well-being[5–7] such as increased occurrences of depression,[8] social isolation,[7] poor social interactions,[9–11] and cognitive dysfunction,[12] as well as increased risk of developing dementia.[13] In addition the impact on normally hearing significant others (SOs) living with hearing-impaired people, is often overlooked where similar problems are reported.[9 14–17]

In spite of robust evidence that hearing aid (HA) use reduces the negative impact of age-related hearing impairment on quality-of-life,[5 18–20] it is estimated that up to

30% of UK adult HA owners do not use them regularly or at all.[3 21–23] The reasons behind this are complex.[24] Numerous barriers and facilitators to successful HA use have been identified as being related to expectations of benefit and meaningful participation in everyday life.[25 26]

The Active Communication Education (ACE) programme[27] trains participants to develop solutions to specific difficult communication scenarios that commonly lead to their avoidance of, or reduced participation in, daily activities. Three small studies have demonstrated benefits in improving communication function and hearing related quality of life when evaluated as an alternative to HA fitting.[28–30] However, the effect of ACE on improving HA benefit when delivered in addition to HA fitting has not been evaluated. The ACE To HEAR study (ACE To improve HEARing) aimed to assess whether a large randomised controlled trial (RCT) to evaluate the effectiveness and cost-effectiveness of ACE plus treatment-as-usual (TAU) versus TAU alone, was feasible within UK National Health Service (NHS) audiology departments.

## DESIGN AND METHODS

The methods are summarised below and published in more detail elsewhere.[31] This was an open feasibility RCT with embedded economic and process evaluations. The required regulatory approvals were in place prior to study commencement.

### Setting

The participating centres were York Hospital, York Teaching Hospitals NHS Foundation Trust, and Bradford Royal Infirmary, Bradford Teaching Hospitals NHS Foundation Trust.

### Intervention

TAU was defined as a referral from a general practitioner (GP) to audiology services to treat permanent hearing loss. It comprised up to two appointments for HA fitting and a third face-to-face or telephone follow-up appointment. The ACE To HEAR intervention consisted of five group sessions delivered as described in the published ACE manual[27] by a trained audiologist facilitator to groups of 5–7 HA users and their SOs, where possible.

### Study objectives

The study objectives were established around evaluating the delivery of the ACE intervention and the feasibility of trial delivery to assess:

► Trial up-take rates, eligibility, and acceptability of clinic location.
► Recruitment rates, the randomisation process and time to accrue ACE groups.
► ACE attendance and retention among participants randomised to the intervention group.

► Acceptability of ACE with participants, SOs and audiologists.
► Capability, capacity and willingness of audiology departments to support delivery of ACE within existing services;
► Intervention fidelity of delivering ACE.
► The acceptability of study processes to participants, SOs and audiologists.
► Patient-reported outcome measures (PROMs) including piloting a bespoke resource use questionnaire and estimate parameters to be used when designing full-scale RCT.

### Sample size and randomisation

Since this was a feasibility trial, no formal power calculation was conducted. We aimed to recruit 88 HA users aged 18 years or over receiving TAU in the two participating centres, along with their SOs. The full eligibility criteria can be found in the published protocol.[31] The study sites estimated that there would be approximately 400 new HA fitted across the study centres each month. Assuming 10% eligibility, 35% did not attend (DNA) and 30% consent rate, around 350 cases per month across the two centres would have needed to be screened to recruit the required 88 HA users in 12 months.

In the initial months of the trial (April to August 2017), eligibility criteria had included Q1 and Q2 from the International Outcomes Inventory for HA (IOI-HA).[32] Q1 required self-reported HA use of less than 3 hours a day and Q2 required less than moderate benefit for eligibility. However, early scrutiny of the Q1 recruitment data showed evidence of over reported HA use, including a variable presence of confirmatory HA log data and it was therefore removed as a criterion. Q2 was also reviewed and considered too stringent; eligibility was changed to include *moderate* as well as less than moderate benefit as these HA users were also considered to be struggling. These changes were implemented as of 29 August 2017.

Eligible, consenting HA users from the same study site who had completed baseline assessments were to be randomised 1:1 by a remote, centralised randomisation service (provided by York Trials Unit) in batches of 10–14 using block randomisation in a single large block per batch. This was to allow groups of 5–7 to be formed for the ACE intervention sessions. The target was to randomise participants within 3 weeks of consent.

### Measurements

Data collected from participants at baseline via self-completed questionnaires were: IOI-HA;[32] Self-Assessment of Communication (SAC)[33]; EuroQol-5 Dimensions (5-level) Questionnaire (EQ-5D-5L)[34 35] and Short-Form 36 (SF-36),[36] resource use and demographics. SOs completed baseline questionnaires collecting: demographics; IOI-HA: version for SOs (IOI-HA-SO)[37] and Significant Other Scale for Hearing Disability (SOS-HEAR).[14]

**Table 1** Stop: go criteria

| Criteria | Measure |
| --- | --- |
| Seventy per cent of recruitment targets attained for all research components | Numbers referred; numbers struggling with their hearing aid; no of exclusions. |
| Study consent/retention rates and proposed sample sizes, indicate delivery of the full RCT is plausible within a 5-year study period | Numbers screening; numbers consenting; numbers who declined to participate and reasons for declining; numbers withdrawing. |
| Ninety per cent of ACE groups of five to seven consented participants formed within the intervention window with participants attending three of five sessions | Time taken to recruit and logistics of recruiting an optimally sized and located ACE group; time ACE started after randomisation (ACE intervention window). Number of consented participants who failed to attend ACE sessions; no who missed ACE intervention window (ie, unable to attend an ACE group within 1–3 weeks after randomisation). |
| Economic, acceptability, outcome measure and fidelity evaluation data successfully collect | Amount of missing data, although measures with over 10% missing data may be modified/replaced prior to the main trial. |
| Participants, SOs and audiologists evaluate acceptability of the ACE and RCT positively | Number given an appointment for an ACE group session; number of consented participants who failed to attend ACE sessions; facilitator's adherence to the ACE protocol (fidelity); participant and SO thoughts regarding the study (interviews). |

ACE, active communication education; RCT, randomised controlled trial; SOs, significant others.

Those in the ACE group also completed a questionnaire at the end of ACE session five. Measures were IOI-HA and the IOI for Alternative Interventions (IOI-AI).[38] Their SOs completed IOI-HA-SO, SOS-HEAR, and the IOI-AI-SO.[37]

The TAU participants were mailed questionnaires containing the IOI-HA at the equivalent time point; with their SOs completed IOI-HA-SO and SOS-HEAR. Participants and their SOs were asked additional questions regarding their thoughts on the programme (ACE arm only) and being approached to take part in the study (both arms). The facilitator collected data during all five ACE sessions in relation to participants' attendance at sessions, SO attendance and their relationship to participant.

At 3 months postrandomisation both groups (ACE and TAU) and their SOs completed the same measures as at baseline, and those in the ACE group and their SOs also completed the IOI-AI and associated IOI-AI-SO.

The feasibility and the potential for a future large-scale study was to be determined by a set of stop: go criteria (table 1).

### Statistical analysis

A single analysis was conducted at the end of the trial and undertaken in Stata V.15. Baseline and outcome data for HA users are summarised overall and by randomised group, and are reported as mean, SD, median, minimum and maximum for continuous data, and counts and percentages for categorical data. The amount of missing data for each variable is also reported.

The recruitment rate is summarised by month, overall, and by site. Attendance at the ACE sessions by participants and their SOs are detailed.

Questionnaire return rates at each time point are presented overall and by group. It was planned that SDs would be presented for PROMs with 80% CIs to inform future sample size calculations. An analysis to investigate sensitivity of the measures to change, by calculating the standardised response mean, was also planned.

The number of participants withdrawing from the ACE intervention and/or the trial and any reasons for withdrawal are summarised.

### Economic analysis and quality of life data

The present trial aimed to determine the feasibility of conducting an economic evaluation of the ACE to HEAR intervention taking a broad perspective accounting not only for NHS costs but also those observed by HA users. In the UK, National Institute for Health and Care Excellence (NICE) advocates the use of the EQ-5D for measuring quality of life; however, the EQ-5D is not always the most sensitive tool for particular populations for whom the majority of its dimensions may not apply.[39] In a US analysis of a similar population, the SF-36 was used to generate QALYs rather than the EQ-5D, with the SF-36 showing a reduction in the cost per QALY in favour of the intervention.[28] Given the UK preference for the EQ-5D, and the fact that it is shorter, thus less onerous to complete, participants in the feasibility trial were asked to complete both instruments to allow for comparison. Completion rates and mean scores for both instruments are summarised.

A bespoke resource use questionnaire was also designed to estimate the potential resource implications of the intervention versus usual care. Completion rates and mean resource use by item are summarised.

### Process evaluation

The number of ACE sessions delivered was recorded and, to assess fidelity of delivering ACE, the facilitator completed a self-monitoring form, scoring adherence to the ACE manual for each session.

Following delivery of all the sessions, the facilitator was interviewed to explore the training and implementation process and their experiences of delivering ACE including barriers/facilitators to adhering to the ACE protocol. In addition, three audiologists were interviewed, exploring the capability, capacity and willingness of their audiology departments to support the ACE study within their existing services. The acceptability of study processes were also explored.

Three participants in the ACE intervention group, plus one SO, were interviewed after the completion of the ACE sessions. The semistructured interviews explored the acceptability of the ACE (eg, venue, timing, content), its perceived impact (reflecting on HA outcomes) as well as exploration of participant's views on study processes (eg, recruitment, outcome measures and timing).

Interviews were audiorecorded, transcribed verbatim and analysed (with NVivo V.11), using the Framework approach,[40] and then interpreted by team members.

## Patient and public involvement

The development of this research study was informed by a variety of patient and public involvement (PPI) activities. A funded public engagement event about public perceptions of hearing impairment helped inform the research question as participants identified a need for wider availability of treatments in addition to HA especially non-technological interventions, such as communication education. This feedback led to the choice of an interactive communication-based intervention rather than an informational one and reinforced the need to ensure that routine practical information about HA and hearing impairment is delivered consistently and checked after fitting. A focus group was then held to consult with service users on the proposed research question, study design and intervention delivery including study information and consent procedures, barriers and facilitators to participation, communication scenarios and motivating factors to becoming more active communicators. In addition, the charity Hearing Link, who have extensive experience of PPI and managing and delivering group interventions of this type, were consulted about involving public and patients in operationalising and delivering ACE. The formation of a project advisory panel consisting of service users ensured PPI involvement continued during the conduct of this study. Patients were not directly involved in recruitment to the study. A lay summary of the findings will be sent to the participants as well as disseminated via patient forums, Trust bulletins and public interest groups.

## RESULTS

### Recruitment

The flow of participants through the trial is shown in a Consolidated Standards of Reporting Trials diagram (figure 1). Screening and recruitment occurred from 1 April 2017 to the end of June 2018; 466 HA users were screened while attending a follow-up HA fitting appointment at Bradford (n=275) or contacted by telephone 3 months postfitting at York (n=191). Of these, 86 (18.5%) were eligible but 74 declined to participate. Twelve participants were recruited into the trial (14.0% of those eligible); 1 from Bradford and 11 from York (online supplemental figure 1 in online supplemental material 1). Reasons for ineligibility (n=380) and non-consent (n=74) can be found in figure 1. The most common reason was that the HA helped too much (81.8% of n=380 excluded). Six SOs also consented to take part.

It was found that 22.6% of HA users who were screened (n=380) were eligible (based on the applicable screening criteria at the time of screening; additional participants were identified as 'would have been eligible' when the inclusion criteria were changed); however, on average only 33 HA users were screened each month, and this was confounded by the 13.9% consent rate (% of the 86 eligible, 8.9% when including those who would have been eligible under the new criteria).

### Randomisation

Randomisation was intended to be performed in batches of 10–14 participants. However, due to slow recruitment, the decision was made in January 2018 to randomise the participants recruited from York thus far as a smaller block to allow at least one ACE group to be run. The randomisation was undertaken as a block of eight, four participants to each group. Additionally, four SOs were also recruited, and allocated to their respective participant's group—three to TAU and one to the intervention group. There were insufficient numbers of participants to randomise the four subsequently recruited participants, and the two SOs.

The average time from screening to randomisation was 19.5 weeks (range 10.6–33.6 weeks) and the target 3-week window for randomisation post-consent was missed for all participants. If the group had been randomised immediately after the eighth participant was consented (randomisation was delayed in the hope of a larger group), then the average time would have been 8.9 weeks, with 37.5% meeting the 3-week target. The first ACE session was conducted 11 days after randomisation.

### Participants

The baseline characteristics of randomised participants and their SOs can be found in table 2. As there was only one SO in the intervention group, data are only presented overall to prevent disclosure.

Online supplemental table 1 in the online supplemental material 1 shows the screening data on use and benefit of HA, threshold data and hearing loss aetiology for the randomised participants by group and overall.

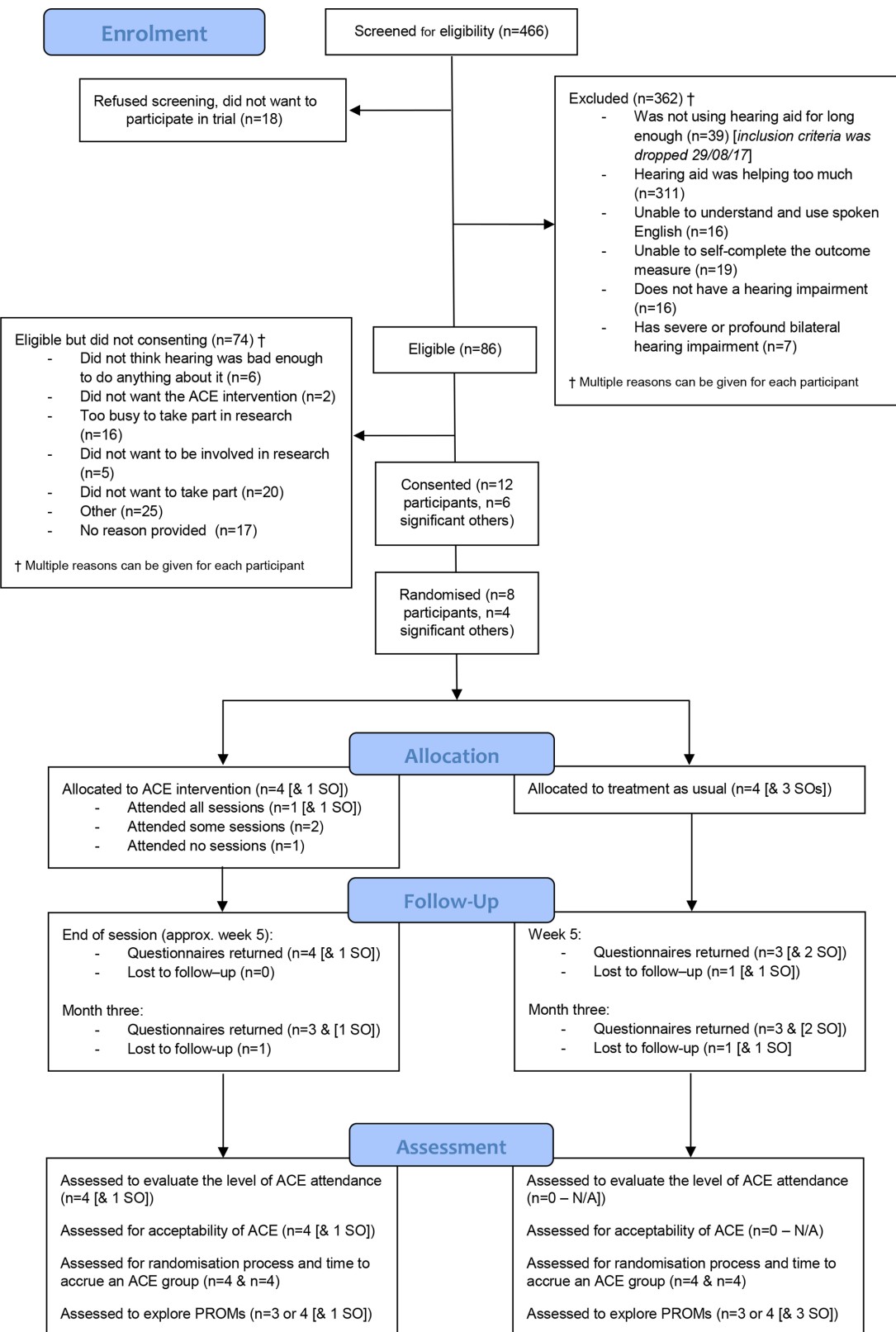

**Figure 1** The flow of participants through the trial shown in a CONSORT diagram. ACE, active communication education; CONSORT, Consolidated Standards of Reporting Trials; PROMs, patient-reported outcome measures; SO, significant other.

Participants had mild to moderately severe high frequency sensorineural hearing loss, except for one who reported an additional conductive element to their hearing loss. Five participants reported using their HA for less than 4 hours a day and three for more than 8 hours a day. All participants reported less than moderate levels of benefit. Those randomised to the ACE intervention reported that they had been concerned about their

**Table 2** Baseline characteristics for the randomised participants, by group and overall and overall for the SOs

| | ACE intervention participants (n=4) | TAU participants (n=4) | All participants (n=8) | All SOs (n=4) |
|---|---|---|---|---|
| **Age, years** | | | | |
| Mean (SD) | 73.0 (7.8) | 73.3 (6.0) | 73.1 (6.4) | 73.6 (7.3) |
| Median (min, max) | 72.9 (64.1, 82.1) | 73.9 (65.9, 79.5) | 73.6 (64.1, 82.1) | 76.6 (62.9, 78.3) |
| **Gender, n (%)** | | | | |
| Male | 3 (75.0) | 2 (50.0) | 5 (62.5) | 2 (50.0) |
| Female | 1 (25.0) | 2 (50.0) | 3 (37.5) | 2 (50.0) |
| **Length of time concerned about hearing, years** | | | | |
| Mean (SD) | 21.3 (7.5) | 3.5 (1.9) | 12.4 (10.8) | N/A |
| Median (min, max) | 20 (15, 30) | 4 (1, 5) | 10 (1, 30) | N/A |
| **Ethnicity, n (%)** | | | | |
| White British | 4 (100.0) | 4 (100.0) | 8 (100.0) | 4 (100.0) |
| Other | 0 (0.0) | 0 (0.0) | 0 (0.0) | 0 (0.0) |
| **Marital status, n (%)** | | | | |
| Married/civil partner | 2 (50.0) | 3 (75.0) | 5 (62.5) | 4 (100.0) |
| Separated | 0 (0.0) | 0 (0.0) | 0 (0.0) | 0 (0.0) |
| Divorced | 1 (25.0) | 1 (25.0) | 2 (25.0) | 0 (0.0) |
| Missing | 1 (25.0) | 0 (0.0) | 1 (12.5) | 0 (0.0) |
| **Living arrangements, n (%)** | | | | |
| Owned outright | 3 (75.0) | 4 (100.0) | 7 (87.0) | 4 (100.0) |
| Owned with mortgage | 0 (0.0) | 0 (0.0) | 0 (0.0) | 0 (0.0) |
| Rented | 0 (0.0) | 0 (0.0) | 0 (0.0) | 0 (0.0) |
| Privately rented | 1 (25.0) | 0 (0.0) | 1 (12.5) | 0 (0.0) |
| **Main activity, n (%)** | | | | |
| Full-time employment | 0 (0.0) | 0 (0.0) | 0 (0.0) | 0 (0.0) |
| Part time | 0 (0.0) | 0 (0.0) | 0 (0.0) | 0 (0.0) |
| Self-employed | 0 (0.0) | 0 (0.0) | 0 (0.0) | 0 (0.0) |
| Unable to work due to health | 1 (25.0) | 0 (0.0) | 1 (12.5) | 0 (0.0) |
| Retired | 2 (50.0) | 4 (100.0) | 6 (75.0) | 4 (100.0) |
| Student | 0 (0.0) | 0 (0.0) | 0 (0.0) | 0 (0.0) |
| Retired and self-employed | 1 (25.0) | 0 (0.0) | 1 (12.5) | 0 (0.0) |
| **Qualifications, n (%)** | | | | |
| No formal qualifications | 2 (50.0) | 1 (25.0) | 3 (37.5) | 0 (0.0) |
| GCSEs | 1 (25.0) | 1 (25.0) | 2 (25.0) | 1 (25.0) |
| A/AS levels | 0 (0.0) | 0 (0.0) | 0 (0.0) | 0 (0.0) |
| Higher education | 0 (0.0) | 1 (25.0) | 1 (12.5) | 2 (50.0) |
| Further higher education | 0 (0.0) | 1 (25.0) | 1 (12.5) | 0 (0.0) |
| Vocational | 0 (0.0) | 0 (0.0) | 0 (0.0) | 0 (0.0) |
| Other | 1 (25.0) | 0 (0.0) | 1 (12.5) | 1 (12.5) |

A/AS, Advanced/ Advanced Subsidiary; ACE, active communication education; GCSE, General Certificate of Secondary Education; SOs, significant others; TAU, treatment-as-usual.

hearing for markedly more years than those allocated to TAU.

## Intervention attendance

Of the four York participants who were randomised to receive the intervention, one attended all five sessions, one attended four sessions, one three sessions and one participant did not attend any of the five sessions. This equates to an average attendance rate of 60%. Of these four participants, only one had an SO who wished to participate, and they attended all the sessions that their associated participant attended.

## Follow-up and standardised measures

At the end of the sessions (approximately week 5), 7/8 participants (87.5%) returned the postal questionnaire (ACE 4/4 (100%); TAU 3/4 (75%)). In addition, 3/4

(75%) SOs returned their questionnaires (ACE 1/1 (100%); TAU 2/3 (67%)).

At 3 months postrandomisation, the retention rate was 75%; three out of four participants from each group returned the questionnaires. Three of the four (75%) SOs returned their 3 months postrandomisation questionnaires. No participants withdrew in this study (but one participant in the ACE group did not attend a single session).

The results from online supplemental table 1 (online supplemental material 1), completed by the participants at both baseline and at least one of the follow-up time points, show that the measures used were well completed. The average IOI-HA score at all three time points was in the mid-to-high range, indicating average hearing outcomes; there was no obvious change in this result in either group over time. There does appear to be imbalance in the SAC between TAU and ACE participants, with those in the control group rating themselves as being worse at communicating. However, as with the IOI-HA there appears to be no change in the average score across the follow-up period.

The outcome measures for the SOs were equally well completed (online supplemental table 1: online supplemental material 1). Since only one SO completed the IOI-AI-SO, the results are not presented.

Out of all of the returned questionnaires (n=8, 6 and 6 for participants and n=4, 3 and 3 for SOs at baseline, W5 and M6, respectively) there was only one instance where a PROM was not completed sufficiently to allow for scoring (IOI-AI for an ACE participant at M6).

The 80% CIs for SDs for PROMs were not calculated, and the planned analysis to investigate the sensitivity of the measures to change was not conducted as it was considered they would not produce reliable results due to the low sample size.

## Health economics

Of the eight participants at baseline, all completed the EQ-5D-5L in full, compared with six full completions (75%) and two partial completions for the SF-36 (25%). At 3 months postrandomisation, two participants (one from each group) did not complete the questionnaire; all remaining participants completed both the EQ-5D-5L and SF-36 in full. At baseline, participants allocated to the intervention group had a slightly higher mean score compared with participants in the TAU group (mean difference 0.057) (table 3). This difference was reversed at 3 months with participants allocated to the TAU group having a slightly higher mean score (mean difference −0.186); although as

the sample size is small, no conclusions can be drawn from this.

These findings were also reflected in the results of the SF-36 (table 4). At baseline, participants allocated to the intervention group scored on average higher on dimensions of physical role, general health, social function, emotional role and mental health as well as on the overall physical and mental components. At the 3 months postrandomisation follow-up, participants allocated to the intervention group only scored higher on the measure of vitality.

The resource use questionnaire was not well completed, with one full completion (12.5%) and five partial completions (62.5%) at baseline and six partial completions only at 3 months postrandomisation (75%). There was no difference in completion by trial group. Overall, very low levels of resource use were reported across both groups with most resource use centred on hearing healthcare professionals (online supplemental table 1: online supplemental material 1).

Appropriate unit costs to be applied for each resource use type were identified through local costings and national databases including Personal Social Services Research Unit[41] and NHS reference costs.[42]

## Adverse events
None were recorded.

## Process evaluation
Five ACE sessions were delivered. At the first session, (1) introduction to ACE and communication needs analysis, the group decided which modules they wanted to use, since each addressed a different aspect of communication. They chose: conversation in background noise (session 2); conversation around the house (session 3); communication with difficult speakers (session 4) and public address systems (session 5). Three HA users and one SO attended.

## Facilitator views and fidelity
Overall, the facilitator considered the ACE programme to have had a positive impact because some of the content was new knowledge for participants, and a few had reported implementing some communication strategies with success.

'The information I was able to give I think was really positive and helped them a lot. I think having spoken to them the following week they'd been able to implement some of them (communication strategies) and found those quite useful. Particularly things like watching the TV at home, that they could put the subtitles on or could

**Table 3** EQ-5D-5L mean (SD) scores and mean difference at baseline and 3 months postrandomisation follow-up

| Follow-up | ACE mean (SD) | TAU mean (SD) | Mean difference (ACE-TAU) |
|---|---|---|---|
| Baseline | 0.670 (0.333) | 0.613 (0.512) | 0.057 |
| 3 months postrandomisation | 0.686 (0.411) | 0.871 (0.133) | −0.186 |

ACE, active communication education; TAU, treatment-as-usual.

**Table 4** SF-36 mean (SD) scores and mean difference at baseline and 3 months postrandomisation follow-up

| SF-36 score | ACE mean (SD) | TAU mean (SD) | Mean difference (ACE - TAU) |
|---|---|---|---|
| **Baseline** | | | |
| Physical functioning | 61.67 (28.77) | 66.25 (44.60) | −4.58 |
| Physical role | 54.69 (40.63) | 48.44 (38.32) | 6.25 |
| Bodily pain | 59 (32.19) | 65.33 (46.88) | −6.33 |
| General health | 61.5 (36.30) | 49.75 (24.31) | 11.75 |
| Vitality | 45.31 (22.46) | 52.08 (15.73) | −6.77 |
| Social functioning | 68.75 (33.07) | 58.33 (50.52) | 10.42 |
| Emotional role | 77.08 (26.68) | 77.08 (15.77) | 0 |
| Mental health | 68.75 (23.94) | 63.33 (12.58) | 5.42 |
| Overall physical component | 41.11 (14.69) | 39.44 (24.22) | 1.67 |
| Overall mental component | 46.87 (10.77) | 45.86 (2.31) | 1.01 |
| **3 months postrandomisation** | | | |
| Physical functioning | 45 (39.05) | 83.33 (10.41) | −38.33 |
| Physical role | 41.67 (38.19) | 68.75 (22.53) | −27.08 |
| Bodily pain | 66.67 (57.74) | 70 (6.93) | −3.33 |
| General health | 58 (41.87) | 58.67 (15.28) | −0.67 |
| Vitality | 54.17 (36.62) | 52.08 (3.61) | 2.08 |
| Social functioning | 62.5 (45.07) | 87.5 (21.65) | −25 |
| Emotional role | 75 (25.0) | 88.89 (19.25) | −13.89 |
| Mental health | 78.33 (20.21) | 78.33 (16.07) | 0 |
| Overall physical component | 36.10 (21.08) | 45.84 (7.134) | −9.74 |
| Overall mental component | 51.33 (10.23) | 51.67 (11.97) | −0.34 |

ACE, active communication education; SF-36, Short-Form 36; TAU, treatment-as-usual.

change the sound settings or that kind of thing. I think they all felt a bit more empowered by the end of it, that they were able to admit that they had a hearing loss and admit that things weren't going to be perfect in every situation.'

Key challenges to its delivery were low numbers of participants which limited discussion, some reluctance among participants to 'bother others' to help with communication, and participants wanting to discuss HA technology and their frustrations with audiology services (instead of communication). As the sessions progressed, the facilitator became more flexible in delivering the programme, ensuring that these other topics were covered. This was reflected in the fidelity scores which changed from 'adherence to the protocol' all the time (session 1), most of the time (sessions 2 and 3) and sometimes (sessions 4 and 5).

**Participants' views**

In reflecting on the ACE programme, two participants and one SO gave several examples of how it was helpful. They had tried out different strategies: sitting with their back to the wall, adjusting settings on their TV, using subtitles and checking if the loop system is switched on when visiting theatres. A key change was increased confidence in speaking out when they cannot hear and asking

others to help with this, for example; facing them when in conversation, requesting people to repeat themselves rather than speak more loudly and closing a window to reduce background noise.

'It also brought it home to me how important it is to tell other people that you've got a hearing problem, because you're reluctant to do it when you're first deaf. And so you muddle along and you say 'yes, yes', and then sometimes you know that they're getting annoyed so then you pretend to hear what was said and of course you laugh in the wrong place or you can't join in the jokes. I always miss punchlines. So, I learned that you must say to people 'I'm sorry, I can't hear, can I sit here? Can we do this? Can we turn that off?''

That said, there was still some reluctance to ask others for help, with participants favouring strategies that they could do themselves. One participant said he felt the training 'wasn't right for me' and that the sessions had not changed anything for him.

**Audiologists' views**

The audiologists said they had been able to deliver the study within their departments; adapting their routine follow-up process to include the ACE screening questionnaire. However, the study was more time consuming than anticipated and requiring extra support from the

university team to help answer questions from HA users was mentioned in one department. Suggested reasons for the failure to recruit were: too strict eligibility criteria (which were changed); a reluctance of older people to participate in research; communication barriers are already discussed in routine appointments; and many HA users nowadays are younger and want to focus on the benefits that technology afford. Ideas for improving recruitment were: to recruit people at their initial fitting, or people who are long term HA users; offer more days and locations for follow-up appointments so more people attend; present ACE as a support group rather than a research study and send flyers out or invite potential participants in for a coffee morning to discuss the study.

Finally, in discussing trial processes, the facilitator and participants observed it had taken a long time to set up the first ACE group and that the information needed to be clearer that the focus of the programme was communication. Participants and audiologists mentioned the length and complexity of the ACE questionnaires. It was difficult to deliver these over the telephone with some elderly HA users.

## DISCUSSION

This is the first study to explore the potential of ACE as a tool to improve communication and HA benefit for struggling adult HA users and their SOs. It is also the first study to identify and evaluate the processes needed to deliver an RCT of ACE within the GP direct referral pathway for new NHS HA users in the UK (TAU). The key findings from this feasibility study are: (1) there is some (limited data) to suggest the intervention is potentially acceptable and feasible to HA users, SOs and audiologists; (2) the data show that the data collection, monitoring and reporting aspects of the trial delivery were feasible and acceptable; finally, (3) we were unable to demonstrate the efficacy of recruitment to the trial within a NHS setting. We did, however, identify some problems that underpin the difficulty of recruitment in this context.

Recruitment to RCTs is challenging and it is important to report the root causes of recruitment difficulties so that mistakes are not repeated in future studies. Emerging data were explored by the trial team to consider how to problem solve and optimise recruitment and associated procedures.

The number of patients fitted with an HA who were followed up and screened for eligibility (n=466), was extremely low in comparison to the number of expected HA fittings (>4000). The study sites' data reporting mechanisms struggled to accurately report the patient flow (monthly HA fittings) through this pathway but actual non-attendance rates for patients offered a follow-up appointment were calculated to be 30%, the level planned for, and do not account for this short fall.

During the study planning phase, both sites agreed to introduce face-to-face or telephone follow-up (not previously offered, despite NICE guidelines indicating this

as good practice[43]) for all patients in the TAU pathway. The low number of patients seen for follow-up and then screened for eligibility by both sites suggests fully embedding and scaling-up this change took longer in practice than anticipated. Not all patients in the pathway were offered post-fitting follow-up and not all who attended were screened, leading to a significantly smaller pool of potential participants to screen (problems also highlighted by Donovan et al[44]).

There is some evidence that RCTs with more restrictive eligibility criteria exhibit poorer recruitment than those with wider criteria.[45] After 6 weeks of missed monthly recruitment targets, we reviewed our eligibility criteria. As noted previously, the relationship between daily hours of HA use and HA outcomes is not well specified.[46] Consequently, criterion Q1 (hours of use), was dropped and the level of benefit (Q2) criterion widened to include patients reporting moderate benefit. Audiologists' feedback supported these changes and by the end of recruitment over 26% of those screened had identified as struggling. We had factored in a conservative estimate of 10%, based on clinical experience at each study site. The disparity reflects the modified eligibility criteria and variability in self-reported HA use data noted earlier.

Despite these problems, the screening process appeared to successfully identify the target population since the demographic profile of those screened and consented was as expected. The range of ages (64.1–82.1) indicated that it was older, not younger participants who were both eligible and willing to participant. Their hearing profile was consistent with that expected from struggling HA users.

Our a priori estimate (30%) of willing participants was closer to 14%. Since willingness is dependent on project specifics, population demographics and patient perceptions of the research's value and worth, this was a feasibility parameter of interest. We could find no relevant study reporting detailed recruitment statistics in this context for comparison.

Comparing our recruitment issues with earlier ACE studies[27 29] is difficult since their context was very different, not least because they recruited from an already motivated population, rather than a specific publicly funded pathway known to foster motivational challenges, as reported here. Descriptions of the burden associated with RCT participation include, anxiety about taking part in research, factors related to processing trial information, demanding follow-up and lack of appropriate post-trial management.[47] Our patients were possibly already feeling overwhelmed by managing their new HA and were less motivated to engage in research. Of the eligible, non-consenting patients, the majority gave no reason, were not interested or were too busy. Hickson et al[28] reported a much lower non-consent rate (20% compared with 86% here) and none of the burdens identified by Naidoo et al[47] were overtly apparent.

It is notable that patients' decisions to not take part were made at clinical follow-up; few were passed on to

the research nurse for further information and consent. Audiologists reported struggling to find time to answer participants' questions fully (either by phone or face to face). Thus, the opportunity for prospective participants to process the trial information at this point may not have been optimum, and they declined referral to the Research Nurse. However, this was not reflected in the patients' narrative on non-consenting.

Research involvement can also be burdensome for clinicians and time is a frequently voiced barrier to their participation in research.[48 49] Pressures from usual clinical practice,[48 50] time demands of recruitment and follow-up, leading to increases in workload pressure[51] are associated with under-recruitment, as is a lack of research experience,[48 52] which was something the interviews with the screening audiologists revealed as a critical concern.

During the project development, audiologists and service managers said they could adapt their practice to include ACE screening. However, over time, they considered the screening questionnaire to be onerous and one site felt that they required more help from the 'university team' to answer patient's questions, despite having a dedicated research nurse/practitioner on site.

It is unlikely that patients declined to participate because they thought the research unimportant since participant interviews and prestudy PPI work indicated our research question was perceived as important.

Important aspects of recruitment such as ensuring simple procedures for information provision and consenting, and that study-related follow-ups do not increase pressure on usual clinical follow-ups (either for clinicians or patients)[52 53] were not really tested here because so few participants consented. Interviews with those who did indicated these aspects were not off-putting.

Many aspects of the study processes worked well. The randomisation process was viewed as acceptable by the small cohort recruited although insufficient numbers were achieved to allocate the last four participants recruited. There was no sense from the recruitment process that SO recruitment was problematic or that including SOs had a detrimental effect on recruitment. The SO/participant ratio (although from minimal data) is commensurate with similar studies[15 28] and was viewed as a positive incentive by participants. Having an SO was not an inclusion criterion, not disadvantaging those without an SO.

The trial processes undertaken for collecting outcome measures worked well. Outcome measure completion rates and interview data suggests that processes worked and are acceptable to participants. Due to insufficient participant numbers, estimation of SD for the PROMS, quality-of-life tools and healthcare resource use questionnaires was not possible. Completion and retention rates both indicate that while we were unable to recruit into the trial, those who were randomised stayed involved, suggesting a trial may have been possible had recruitment not been an issue. Attempts to increase recruitment via amendments to streamline recruitment paperwork and processes failed, and further study site training proved

unsuccessful. We also planned to (1) add additional study sites; (2) change the point in the referral pathway at which HA users were approached, to the fitting appointment, rather than the 3 months follow-up appointment and (3) to transfer all responsibility for recruitment from the NHS audiologists to the university research team. Ethical approval was gained for these adjustments and the funder granted the additional time required, but refused additional funding.

The ACE delivery processes appeared to work well, with the facilitator becoming more flexible over the five sessions to accommodate participant's needs within the ACE protocol. Participants mostly completed the programme and were generally positive in their feedback, citing changes they were making. This is in line with previous studies using ACE in Australia, Sweden and Chile.[28 30 54] All previous studies reported good adherence to the ACE programme (>90% completed three or more of the five sessions), and positive outcomes were indicated in terms of communication. All studies found pre-ACE and post-ACE improvements in communication strategy use, particularly for those with mild hearing loss; activity and participation and psychosocial well-being, with findings reflected in positive feedback. Despite the small numbers of participants in this current study, participants also reported positive benefits from completing the ACE programme. A key challenge to the efficacy of delivering ACE here was the small group sizes that limited group discussion, so it was interesting and encouraging to note the broad changes people made in their communication strategies because the ACE programme.

## CONCLUSION

A future RCT using the recruitment strategy reported here is not likely to be feasible. The fundamental problem was in accessing sufficient struggling HA users at follow-up. Recruiting HA users when first fitted may be a better strategy for future research. Additional work to investigate which point in the pathway patients should best be approached and to identify the resources needed for good recruitment is required. Work to develop a robust service delivery model of follow-up for new adult HA users based on NICE guidelines will be crucial, as would implementing organisational support for NHS audiologists involved in research.

**Author affiliations**
[1]York Trials Unit, Department of Health Sciences, University of York, York, UK
[2]Valid Research Ltd, Wetherby, UK
[3]Leeds Institute of Cardiovascular & Metabolic Medicine, Univerity of Leeds, Leeds, UK
[4]School of Health and Rehabilitation Sciences, The University of Queensland, Brisbane, Queensland, Australia
[5]Academic Unit of Elderly Care and Rehabilitation, University of Leeds, Bradford, UK
[6]Audiology Department, Bradford Royal Infirmary, Bradford, UK
[7]Audiology Department, York Teaching Hospital NHS Foundation Trust, York, UK
[8]Hearing Link, Eastbourne, UK

**Acknowledgements** This work uses data provided by patients and collected by the NHS as part of their care and support and would not have been possible

without access to this data. The Authors would also like to thank the members of the Project Advisory Panel (PAP), Emmanuelle Blondiaux-Ding, Philip Le Mare and Shona Hudson for their invaluable advice on many aspects of the project.

**Contributors** NJT led on the original conception and design of the study with substantial contributions to the protocol from JW, CJ, EC, AF, LH, KB, CF, CH, CM, LG, RG and KI. As chief investigator, NJT takes overall responsibility for the work. CM liaised with the participants, delivered the ACE intervention and oversaw the PAP meetings. JW made substantial contributions to the trial design and management and manuscript drafting. EC was responsible for data analysis and data interpretation. KB was responsible for the health economic design and analysis. CF and CH provided further statistical advice and expertise on the study. CJ was responsible for qualitative interviews and data collection and transcript analysis. AF was involved in all aspects of the study and KI and RG were recruiting site leads, responsible for aspects specific to Trust's service delivery and providing expert clinical support. All authors contributed to draft manuscripts, read and approved the final manuscript.

**Funding** This paper presents independent research funded by the National Institute for Health Research (NIHR) under its Research for Patient Benefit (RfPB) Programme (Grant Reference Number PB-PG-0215-36147).

**Disclaimer** The views expressed are those of the author(s) and not necessarily those of the NHS, the NIHR or the Department of Health and Social Care.

**Competing interests** None declared.

**Patient consent for publication** Not required.

**Provenance and peer review** Not commissioned; externally peer reviewed.

**Data availability statement** Data are available on reasonable request. The data that support the findings of this study are available from the corresponding author, (JW), on reasonable request.

**ORCID iDs**
Judith Watson http://orcid.org/0000-0003-0694-3854
Nicholas J Thyer http://orcid.org/0000-0002-0025-783X

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
