## [Reviewer comments · BMJ Open]

ARTICLE DETAILS

TITLE (PROVISIONAL)	A randomised controlled feasibility trial of an Active Communication Education programme plus hearing aid provision versus hearing aid provision alone (ACE To HEAR)
AUTHORS	Watson, Judith; Coleman, Elizabeth; Jackson, Cath; Bell, Kerry; Maynard, Christina; Hickson, Louise; Forster, Anne; Fairhurst, Caroline; Hewitt, Catherine; Gardner, Rob; Iley, Kate; Gailey, Lorraine; Thyer, Nicholas

VERSION 1 – REVIEW

REVIEWER	Wanderleia Q. Blasca University of São Paulo - Brasil
REVIEW RETURNED	08-Sep-2020

GENERAL COMMENTS	The presented manuscript is of great importance as scientific research, but mainly in clinical practice. Analyzing significant aspects of the hearing aid's adaptation process, it aims, through greater detail, quality of life's improvement. Through the coherence of the objectives, and the research's design emphasizes the proposal's justification and social relevance. THE ARTICLE ENTITLED – A RANDOMISED CONTROLLED FEASIBILITY TRIAL OF AN ACTIVE COMMUNICATION EDUCATION PROGRAMME PLUS HEARING AID PROVISION VERSUS HEARING AID PROVISION ALONE (ACE TO HEAR) Had seeked to establish ACE (Active Communication Education) acceptability and viability on quality of life and development of the communication regarding hearing impaired patients who used hearing aids from the National Health Service (NHS). In this context, your great importance was administrating and bringing the proposal of accompanying and effectiveness of the hearing aid's adaptation process to the clinic of hearing health service. Aiming to lower hearing hardships and, mostly, the hearing aid's abandonment and, consequently, the aggravation of the user's auditory and non-auditory implications. • With this purpose, it was innovative in your objectives and research design;• The manuscript showed itself to be appropriate regarding editorials norms;
--

	 • The methodology was appropriated to the objectives of the study. However, during the realization of the proposal it demonstrated your fragility, in my view, specifically, directed to clinical practice. I believe this to be a great scientific hardship, the adequacy of the best practices the methodologies and clinical protocols that revolve the professional and the patient; • The presentation and adequate interpretation of the results were showed in a clear manner; • The discussion is coherently dialoguing with the proposed goals and obtained results; • The conclusion is based on strong and weak points of the manuscript. Exposing the difficulty in the strategy of recruitment patients as an important factor related to the proposal's small number of participants. Reflecting over new proposals and strategies that could, probably, envelop a bigger number of participants, aiming to a more detailed study regarding communication and quality of life of the hearing impaired patient. I would like to point out, as a suggestion in the conclusion, the emphasis of the positive result of the utilization of the ACE in the hearing aid's adaptation process. Results demonstrated by the patients and, also, by audiologists, valuing the excellency of the research. Finally, based on my evaluation, I suggest the approval of the manuscript for publication.
--	---

REVIEWER	Alex Meibos The University of Akron, United States of America
REVIEW RETURNED	03-Oct-2020

GENERAL COMMENTS	Thank you for the opportunity to review this manuscript. I believe it highlights the greatest challenges that clinical audiology researchers face worldwide, the ability to design and carry out large scale RCTs, and control for the challenge of recruitment and variability in patients' lives and outcomes with hearing technologies and aural rehabilitation. In my experience, every attempt to study group aural rehabilitation or support groups with adults who have hearing loss all end up with small sample sizes and it is For my review checklist, I indicated N/A when answering about whether the methods were described sufficiently to allow the study to be repeated, since the authors pointed out that the methods have already been published in a different manuscript published by BMJ Open in 2018. I also checked N/A when answering if the supplementary reporting was complete (e.g. trial registration; funding details; CONSORT, STROBE or PRISMA checklist). As far as I could see, it was, but I am not familiar with these reporting forms, so, I did not feel qualified to establish any comment or opinion regarding the authors' accuracy and completeness for this item.
---

	The minor revisions I would recommend are comments to revise wording where sentences were not clear to me: Page 22 (23 of 82) - Revise wording of discussion item number 3 to make clearer to readers "(3) we were unable to we demonstrate that is it possible to recruit participants to this trial, suggesting that it is difficult to do within the NHS setting." Revise wording on page 26 (27 of 82) in the last two sentences of the second paragraph to make clearer to readers: "A plan to add more centres, change the point at which HA users were approached to the fitting appointment, and to transfer all responsibility for recruitment from the audiologists to the research team was ethically approval. However, although the funder granted the additional time required, extra funding was refused." Despite the low sample sizes and challenges with recruitment, I think a study like this will help contribute to the audiology literature greatly in identifying the challenges clinical audiology researchers face in the organization and design of larger scale studies.
--	--

REVIEWER	Norbert Dillier University of Zurich, Switzerland
REVIEW RETURNED	05-Oct-2020

GENERAL COMMENTS	This paper describes a well prepared randomized study about possible increased hearing aid provision benefits through structured active communication education intervention. Due to very poor recruitment outcomes only limited conclusions about the effect of the intervention could be drawn. The authors describe in detail the problems during the recruitment of study participants and discuss possible reasons for the very modest final number of only eight subjects, divided into a control and intervention group of 4 subjects each. The frustration about the poor outcome after a long and laborious preparatiion is quite obvious and understandable. Whether the anticipated "increases in workload pressure" and the perceived lack of help from the "University Team" to answer patient's questions can be attributed to local problems or constitutes a more serious deficit related to the British NHS remains open. Similar studies in other countries (Australia, Sweden and Chile) apparently were more successful in this respect. Although the authors conclude that a future (or follow-up) study would not be feasible the publication of this fully documented feasibility trial seems to be worthwhile and interesting for an international audience. Minor comments: P12/L20: interpreted by team members P16/L20: No participants withdrew in this study (but one participant in the ACE group did not attend a single session). P22/L38: we were unable to demonstrate P26/L45: ethically approved
--

REVIEWER	Guangyu Tong Yale University
REVIEW RETURNED	10-Nov-2020

GENERAL COMMENTS	The outstanding issue for this trial is that there are only 8 patients enrolled, preventing this study from obtaining any meaningful results. Even with only 8 patients, there are still missing data issues for some outcome (as described on p.17). It is statistically inappropriate to construct parametric statistics like 95% CI since nothing is empirically normal under such a small sample size.
--

REVIEWER	Irina Chis Ster St George's University of London
REVIEW RETURNED	19-Nov-2020

GENERAL COMMENTS	I appreciate the difficulties the research team had with recruitment. I also understand that the study is a pilot and/or feasibility and not a hypothesis testing. Therefore, I would recommend to take out any 95% interval as independent comparisons between groups of size 4 makes little sense. The authors do not overinterpret the results - however, among applicable tests would be permutations/randomization tests. But tests should not be carried in the first instance as they aren't informative enough.
---

VERSION 1 – AUTHOR RESPONSE

Reviewer 1

No changes are required from this reviewer.

Reviewer 2

Comment: Page 22 (23 of 82): Revise wording of discussion item number 3 to make clearer to readers "(3) we were unable to demonstrate that it is possible to recruit participants to this trial, suggesting that it is difficult to do within the NHS setting."

Response -Thank you for pointing out that this part was unclear. We have reworded it to state: "finally, (3) we were unable to demonstrate the efficacy of recruitment to the trial within an NHS setting. We did, however, identify some problems that underpin the difficulty of recruitment in this context."

Comment: Revise the wording on page 26 (27 of 82) in the last two sentences of the second paragraph "A plan to add more centres, change the point at which HA users were approached to the fitting appointment, and to transfer all responsibility for recruitment from the audiologists to the research team was ethically approved. However, although the funder granted the additional time required, extra funding was refused."

Response -This has now been revised to read: "We also planned to (1) add additional study sites; (2) change the point in the referral pathway at which HA users were approached, to the fitting appointment, rather than the 3-month follow-up appointment; and (3) transfer all responsibility for recruitment from the NHS audiologists to the University research team. Ethical approval was gained for these adjustments and the funder granted the additional time required but refused additional funding."

Reviewer 3

Response -The minor comments from this reviewer have been addressed in the revised text:

P12/L20: interpreted by team members – the word ‘in’ has been removed as pointed out by the reviewer.

P16/L20: No participants withdrew in this study (but one participant in the ACE group did not attend a single session). – we have added the section in brackets as suggested by the reviewer.

The following suggestions have been now incorporated into our response to Reviewer 2’s comments.

P22/L38: we were unable to demonstrate – we have removed the unnecessary extra ‘we’.

P26/L45: ethically approved – in light of comments made by another reviewer, the structure of this sentence has now changed.

Reviewer 4

Comment: It is statistically inappropriate to construct parametric statistics like 95% CI since nothing is empirically normal under such a small sample size.

Response -Thank you for this comment. All 95% confidence intervals (which were found in Tables 3 & 4) have now been removed.

Reviewer 5

Comment: I would recommend to take out any 95% interval as independent comparisons between groups of size 4 makes little sense.

Response -Thank you. In response to a previous reviewer, the 95% confidence intervals (found in Tables 3 & 4) have now been removed.

VERSION 2 – REVIEW

REVIEWER	Wanderleia Blasca Universidade de São Paulo - Brasil
REVIEW RETURNED	02-Jan-2021

GENERAL COMMENTS	The article “A randomized controlled feasibility trial of an Active Communication Education program plus hearing aid provision versus hearing aid provision alone (ACE To HEAR)” was widely analyzed by reviewers who carefully listed the most important suggestions for qualifying the article. In the subsequent reading, it was possible to verify the authors' care in responding carefully to the suggestions made by the reviewers, prioritizing the quality of the scientific article. Thus, I would like to emphasize once again the importance of the work performed demonstrating the difficulties presented in the clinical routine for the care of the hearing impaired patient, as well as their weaknesses, in a context not only of the United Kingdom's National Health Service, but in most specialized services in the field of audiology. Finally, based on my assessment, Approving the manuscript for publication.
--

REVIEWER	Alex Meibos The University of Akron, United States of America
REVIEW RETURNED	28-Dec-2020

GENERAL COMMENTS	The minor revisions I recommended were addressed by the authors well, and I believe they do help improve the clarity of the manuscript in these areas. It appears that other suggestions/recommendations made by all reviewers of this manuscript were also implemented. I recommend this revised manuscript for publication in BMJ open.
---

REVIEWER	Norbert Dillier University of Zurich, Switzerland
REVIEW RETURNED	23-Dec-2020

GENERAL COMMENTS	The additional information in the revised version is helpful to appreciate the efforts done to prepare and carry out this study. My comments on the first version have been fully considered. I have no further comments.
--